# Exploiting the Nucleic Acid Nature of Aptamers for Signal Amplification

**DOI:** 10.3390/bios12110972

**Published:** 2022-11-04

**Authors:** Miriam Jauset-Rubio, Mayreli Ortiz, Ciara K. O’Sullivan

**Affiliations:** 1Interfibio Consolidated Research Group, Department of Chemical Engineering, Universitat Rovira I Virgili, 43007 Tarragona, Spain; 2Institució Catalana de Recerca I Estudis Avançats (ICREA), Passeig Lluís Companys 23, 08010 Barcelona, Spain

**Keywords:** aptamer, aptasensors, biotinylated dNTPs, enzyme-linked oligonucleotide assay (ELONA), chronoamperometry, electrochemical detection

## Abstract

Aptamer-based assays and sensors are garnering increasing interest as alternatives to antibodies, particularly due to their increased flexibility for implementation in alternative assay formats, as they can be employed in assays designed for nucleic acids, such as molecular aptamer beacons or aptamer detection combined with amplification. In this work, we took advantage of the inherent nucleic acid nature of aptamers to enhance sensitivity in a rapid and facile assay format. An aptamer selected against the anaphylactic allergen β-conglutin was used to demonstrate the proof of concept. The aptamer was generated by using biotinylated dUTPs, and the affinity of the modified aptamer as compared to the unmodified aptamer was determined by using surface plasmon resonance to calculate the dissociation constant (K_D_), and no significant improvement in affinity due to the incorporation of the hydrophobic biotin was observed. The modified aptamer was then applied in a colorimetric competitive enzyme-linked oligonucleotide assay, where β-conglutin was immobilized on the wells of a microtiter plate, competing with β-conglutin free in solution for the binding to the aptamer. The limit of detection achieved was 68 pM, demonstrating an improvement in detection limit of three orders of magnitude as compared with the aptamer simply modified with a terminal biotin label. The concept was then exploited by using electrochemical detection and screen-printed electrodes where detection limits of 326 fM and 7.89 fM were obtained with carbon and gold electrodes, respectively. The assay format is generic in nature and can be applied to all aptamers, facilitating an easy and cost-effective means to achieve lower detection limits.

## 1. Introduction

In the last three decades, aptamers, nucleic acid analogues of antibodies with the potential to bind to a wide range of target molecules, have been exploited in a variety of applications, including analytical tools and therapeutics [1]. Aptamers are normally selected by using a method termed Systematic Evolution of Ligands by Exponential enrichment (SELEX), an iterative process that begins with a combinatorial library of nucleic acids, typically of 10^15^ diverse sequences, and through sequential steps of target incubation and partitioning to remove non-target-binding sequences, finally resulting in the identification of target-binding sequences. Whilst there are a plethora of reports detailing the successful selection of high-affinity aptamers, the ability to improve aptamer affinity would effectively enhance their potential for application.

The limited chemical diversity of nucleic acids, comprising four natural nucleotides, as compared to antibodies that can be formed from up to twenty different hydrophilic and hydrophobic amino acids, is a potential limitation of aptamers in binding targets, particularly proteins that may contain both hydrophilic and hydrophobic. Furthermore, aptamers have a hydrophilic polyanionic backbone, limiting interaction with hydrophobic moieties and thus reducing the diversity of potential binding sites [2]. However, as aptamers are inherently nucleic acid in nature, they are thus amenable to diverse strategies for improved target binding.

To address this, alternative strategies, including pre- and post-SELEX modifications, have been pursued. The use of an “extended alphabet” of nucleic acids where artificial hydrophobic nucleotides have been employed in SELEX libraries [3,4] has facilitated the selection of aptamers that are capable of binding to intramolecular motifs, previously inaccessible with “traditional” aptamers, generating molecules with binding affinities in the nM—pM range [5]. These chemical modifications to produce artificial nucleotides can be introduced via the base (e.g., 5-methyluridine, pseudouridine, and dihydrouridine), sugar (e.g., 2′-O-methyl, 2′-Fluoro, 2′-fluoroarabino nucleic acid (2′-FANA), and locked nucleic acid (LNA)), or backbone (e.g., morpholino, peptide nucleic acid, phosphorothioate, and boronophosphate) [6]. An alternative approach is exploited in the slow off-rate modified aptamers (SOMAmers), which are DNA aptamers containing 2′-deoxyuridine nucleotides that have modifications on the C5 position of the base with different protein-like moieties (e.g., benzyl and 2-napthyl or 3-indolyl-carboxamide), as exemplified by the PDGF-BB aptamer [7]. The use of these modified nucleotides is fascinating and will undoubtedly significantly enhance the binding properties of aptamers, allowing accessibility to both hydrophilic and hydrophobic epitopes. However, the use of these modified libraries requires the use of specifically designed and inherently expensive polymerases that are capable of incorporating modified artificial nucleotides as substrates [8].

An alternative approach is to modify the selected aptamers post-SELEX in order to avoid the requirement of specialized enzymes for amplification during the selection process, and indeed, some chemical modifications, such as the use LNA aptamers or SOMAmers, have been reported to be implemented following SELEX [9,10,11]. However, re-screening of the generated modified aptamers is strongly recommended, as these modifications may drastically enhance or even reduce aptamer-binding parameters.

One potential strategy that has the possibility of not only endowing hydrophobic properties to aptamers selected by using natural nucleotides, but also offering an anchor for signal amplification in analytical application of the post-SELEX modified aptamer, is the use of biotinylated nucleotides. Biotin (vitamin H) is a small hydrophobic molecule that functions as a coenzyme of carboxylases, and the avidin–biotin complex is the strongest known non-covalent interaction (K_D_ = 10^−15^ M) between a protein and ligand. The bond formation between biotin and avidin is rapid and is unaffected by extremes of pH, temperature, organic solvents, and other denaturing agents. The first reports of the use of biotin labeling of nucleic acids was in the 1970s, when Davidson’s team used nucleic acid-biotin complexes as probes for in situ hybridization [12,13]. Langer et al., (1981) went on to synthesize the first examples of biotinylated UTPs and TTPs, which were demonstrated to be incorporated by RNA and DNA polymerases [14,15,16]. Subsequently, Gebeyehu et al. reported the synthesis of biotinylated ATP and CTP [17], and since then a variety of methods for the synthesis of biotinylated nucleotides have been reported [18,19,20]. The combined incorporation of all four nucleotides bearing biotin labels has been reported [21,22]. These biotinylated nucleotides have been successfully employed for signal amplification in a variety of applications, including enzyme-catalyzed deposition [23], microarrays [24], detection of microRNAs [25,26], pathogen nucleic acids [27], and single nucleotide polymorphisms [28], as well as in the detection of DNA [29], electrochemical immunosensors [30], and in lateral flow assays [31]. Furthermore, the use of biotinylated nucleotides has also been reported for use in aptamers for signal enhancement [32,33].

In the work reported herein, we take advantage of a DNA aptamer we previously selected against the anaphylactic allergen β-conglutin (β-CBA II aptamer) as a model system to demonstrate a proof of concept of the improved assay sensitivity achieved via the incorporation of biotinylated dUTPs into the aptamer. Whilst we did not carry out SELEX by using a library bearing the biotinylated dUTPs, we evaluated if the post-SELEX introduction of these hydrophobic nucleotides would improve the affinity of the aptamer through potential improved accessibility to hydrophobic regions of the β-conglutin target and, thus, end the affinity of the modified as compared to the non-modified aptamer; this was determined by using surface plasmon resonance. The modified aptamer was then employed in both colorimetric and electrochemical competitive enzyme-linked oligonucleotide assays and the detection limits achieved compared with that of the aptamer linked to a single biotin label. The developed assay is robust, facile, and generic in nature and can theoretically be applied to all aptamers, thus facilitating lower achievable detection limits, which is of particular importance where ultrasensitive detection is required, or for aptamers with relatively low affinities.

## 2. Materials and Methods

### 2.1. Materials

Phosphate-buffered saline (PBS; 10 mM phosphate, 137 mM NaCl, 2.7 mM KCl, pH 7.4), PBS-Tween (10 mM phosphate, 137 mM NaCl, 2.7 mM KCl, 0.05% (*v*/*v*) Tween 20, pH 7.4), magnesium chloride (MgCl_2_), 11-mercaptoundecanoic acid (MUA), skimmed milk powder, sulfuric acid, 3,3′,5,5′-tetramethylbenzidine (TMB), potassium hydroxide (KOH), ethanolamine, 50x Denhardt’s solution, and carbonate–bicarbonate buffer were obtained from Sigma Aldrich (Barcelona, Spain). Lambda exonuclease, GeneRuler Low range DNA ladder, Agarose (Low-Melting, <1kb DNA/RNA/Genetic Analysis Grade), NUNC Maxisorp microtiter plate, 1-Ethyl-3-(3-dimethylaminopropyl) carbodiimide (EDC), N-hydroxysuccinimide (NHS), and all other reagents were purchased from Fischer Scientific (Madrid, Spain). Streptavidin Poly-HRP80 was supplied from SDT-reagents (Baesweiler, Germany). A biotin PCR Labeling Core Kit was acquired from Jena Bioscience (Jena, Germany). A CM5 sensor chip was bought from Cytiva (Madrid, Spain). All DNA oligonucleotides were purchased from Biomers (Germany) and can be found in Appendix A. All solutions were prepared in high-purity water obtained from the Milli-Q RG system (Barcelona, Spain).

### 2.2. Aptamer Preparation

#### 2.2.1. Incorporation of Biotinylated dNTPs

The introduction of biotinylated dNTPs in the amplicon was carried out by using the Polymerase Chain Reaction (PCR) with 50 pM of unmodified aptamer. PCR was performed by using Biotin PCR labeling Core Kit (Jena Bioscience) in a total volume of 50 μL, with 200 nM of primers (unmodified forward primer and phosphorylated reverse primer), 1x PCR labeling buffer, 1x Biotin PCR labeling mix (containing 50% biotinylated dUTP), and 1 U of Taq polymerase. The program used an initial heating step at 95 °C for 2 min, followed by 25 rounds of 30 s denaturation at 95 °C, 30 s annealing at 58 °C, and 30 s elongation at 72 °C, followed by a final extension step at 72 °C for 5 min. Double-stranded PCR products were visualized by using gel electrophoresis via the addition of 5 μL of amplified sample with 4 μL of 6x loading buffer to a 3% (*w*/*v*) agarose gel stained with GelRed nucleic acid stain (VWR, Spain) and imaging with a UV lamp (λ = 254 nm).

#### 2.2.2. Asymmetric Polymerase Chain Reaction (A-PCR)

Ten microliters of the resulting amplicons from PCR was used as template for asymmetric PCR (A-PCR), using the biotin PCR labeling Core Kit. A total volume of 100 μL reaction was prepared with 400 nM of unmodified forward primer, 1x PCR labeling buffer, 1x Biotin PCR labeling mix (containing 50% biotinylated dUTP), and 1 U of Taq polymerase. The program used an initial heating step at 95 °C for 2 min, followed by 12 rounds of PCR, with 30 s of denaturation at 95 °C, 30 s of annealing at 58 °C, and 60 s of elongation at 72 °C, followed by a final extension step at 72 °C for 5 min. The A-PCR products were visualized by using gel electrophoresis.

#### 2.2.3. Enzyme Digestion

Ninety microliters of A-PCR product was incubated with 1x Lambda exonuclease buffer and 10 U of Lamba exonuclease for 90 min, at 37 °C. The reaction was stopped by 10 min of incubation at 80 °C. The modified aptamer was visualized by using gel electrophoresis, as explained above. Finally, the modified aptamer was purified with the Oligo Clean and Concentrator kit (Ecogen, Barcelona, Spain), according to the manufacturer’s instructions, and its concentration was estimated by using the SimplyNano spectrophotometer.

### 2.3. Affinity Studies: Surface Plasmon Resonance (SPR)

Surface plasmon resonance (SPR) was performed by using the BIAcore 3000 (Biacore Inc.). A CM5 sensor chip was activated with EDC/NHS (35 µL of a 1:1 mixture EDC (400 mM) and NHS (100 mM), followed by the injection of 0.2 mg mL^−1^ of β-conglutin in 10 mM Acetate buffer pH 5 at a flow rate of 5 µL min^−1^. Following immobilization of the protein, unreacted NHS esters were deactivated with an excess of ethanolamine hydrochloride (35 µL of 1 M) at the same flow rate. Unbound protein was washed from the surface with repeated injections of 10 µL regeneration buffer (2 M NaCl and 50 mM NaOH) until a plateau was reached. The final immobilization level of β-conglutin was 5487.9 RU. The unmodified and the modified aptamer with biotinylated dUTPs, prepared as described above, were diluted in binding buffer (10 mM phosphate buffer, 135 mM NaCl, 2.5 mM KCl, and 1.5 mM MgCl_2_, pH 7.4), ranging from 125 nM to 0 nM, using 1 in 2 dilutions. Injection of the modified/unmodified aptamer was carried out during 6 min at a flow rate of 5 µL min^−1^, followed by 3 min of stabilization time and 7 min of dissociation time. The binding affinity of the aptamers was determined by BIAevaluation software, using a 1:1 Langmuir binding model. The signal of the specific flow cell (with immobilized β-conglutin) was corrected by the subtraction of the signal from the control flow cell (activated with EDC/NHS and blocked with ethanolamine, as described above, but without β-conglutin immobilization). The signals were also corrected via subtraction of the buffer signal obtained. Duplicates were carried out for all the experiments.

### 2.4. Enzyme Linked Aptamer Assay (ELAA)

The sensitivity of both the aptamer with biotinylated dUTPs incorporated and the aptamer bearing a single biotin label for their cognate β-conglutin target was determined by using an enzyme-linked aptamer assay (ELAA). NUNC MaxiSorp microtiter plates were functionalized by pipetting 50 μL of 20 μg mL^−1^ β-conglutin in 50 mM carbonate-bicarbonate buffer pH 9.6 into each well, and the plates were then incubated for 30 min, at 22 °C, and subsequently washed with 0.05% (*v*/*v*) PBS-Tween-20. A blocking step was carried out by adding 200 μL (per well) of 5% *w*/*v* skimmed milk powder in PBS-tween buffer, which was left to incubate for 30 min, at 22 °C, followed by thorough washing.

#### 2.4.1. Evaluation of Biotinylated Aptamer

A range of concentrations of either the aptamer incorporating biotinylated dUTPs, the aptamer bearing a single biotin label or non-cognate aptamer (50 nM–0 nM) prepared in binding buffer (10 mM phosphate buffer, 135 mM NaCl, 2.5 mM KCl, and 1.5 mM MgCl_2_, pH 7.4) were added to the functionalized plates (50 μL per well) and incubated for 30 min at 22 °C, under shaking conditions. This was followed by a washing step with 0.05% (*v*/*v*) PBS-Tween-20 and subsequent addition of 50 μL of a 1 in 20,000 dilution of 1 mg mL^−1^ streptavidin linked to a polymer containing multiple HRP molecules (SA-polyHRP80) to each well and then incubation for a further 30 min. After a final washing step, the presence of HRP was measured via the addition of 50 μL of TMB substrate, with the enzymatic reaction being stopped via addition of 50 μL 1 M H_2_SO_4_ 5 min later. Moreover, three non-cognate targets (α-conglutin, γ-conglutin, and δ-conglutin) were immobilized, as explained above, and incubated with the aptamer incorporating biotinylated dUTPs. The absorbance was read at 450 nm (SpectraMax 340PC384, bioNova Scientifics S.L., Madrid, Spain). The signals obtained were plotted with GraphPad Prism software and fitted to a sigmoidal 4PL model. The limit of detection (LOD) was defined as the lowest signal obtained (bottom value; no target in solution) plus three times its standard deviation (bottom value + 3x SD bottom value), and the value was interpolated from the fitted curve. Triplicate measurements were performed for each concentration.

#### 2.4.2. Competition Assay

Individual Eppendorf tubes containing 0.4 nM of aptamer and incorporating biotinylated dUTPs/bearing a single biotin label were pre-incubated with serial dilutions of β-conglutin (1000 nM–0 nM) in binding buffer (10 mM phosphate buffer, 135 mM NaCl, 2.5 mM KCl, and 1.5 mM MgCl_2_, pH 7.4) for 30 min, at 22 °C. Following incubation, the wells were washed, and the contents of the pre-incubated tubes were added to the plate (50 μL per well) and incubated for a further 30 min. This was followed by a washing step with 0.05% (*v*/*v*) PBS-Tween, a subsequent addition of 50 μL of a 1 in 20,000 dilution of 1 mg mL^−1^ SA-polyHRP80, to each well and a further incubation for 30 min. After a final washing step, 50 μL of TMB substrate was added into the wells, and after 5 min, the enzymatic reaction was stopped with 50 μL 1 M H_2_SO_4_. The absorbance was read at 450 nm (SpectraMax 340PC384, bioNova Scientifics S.L., Madrid, Spain). The signals obtained were plotted with GraphPad Prism software and fitted to a sigmoidal 4PL model. The limit of detection (LOD) was defined as the highest signal obtained (top value; no target in solution) minus three times its standard deviation (top value + 3x SD top value), and the value was interpolated from the fitted curve. Triplicate measurements were performed for each concentration.

### 2.5. Electrochemical Detection

#### 2.5.1. Instrumentation

Electrochemical measurements were performed on a PC-controlled PGSTAT12 Autolab potentiostat (EcoChemie, The Netherlands) controlled with the General Purpose Electrochemical System (GPES) software (Eco Chemie B.V., Utrecht, The Netherlands). Fast chronoamperometry was used to measure the reduction current from the oxidated TMB by the action of the HRP label by applying two consecutive potential steps at 0 V for 0.005 s and −0.1 V vs. Ag for 0.5 s to each electrode and taking the current readout at the end of the second step.

Two kinds of commercially available screen-printed electrodes (SPEs) from Dropsens were used to carry out the experiments. The carbon SPE configuration (DRP-110) was as follows: working electrode, carbon disk (*φ*  =  4 mm); reference electrode, silver; and counter electrode, carbon. The gold SPE (DRP-250BT) was the working electrode: gold disk (*φ* = 4 mm). The reference electrode was silver, and the counter electrode was platinum.

#### 2.5.2. Functionalization of the Screen-Printed Electrodes

Prior to functionalization, the carbon electrodes were activated by cycling ten times, from 0 to −1.2 V vs. Ag in 0.5 M KOH at 100 mV s^−1^. The gold electrodes were activated by cycling ten times, from 0 to 1.2 V vs. Ag in 0.1 M H_2_SO_4_, at 100 mV s^−1^. Following activation, the SPEs were washed with Milli-Q water and dried with N_2_.

For the functionalization of carbon SPEs, 30 μL of 20 μg mL^−1^ β-conglutin in 50 mM carbonate–bicarbonate buffer, pH 9.6, was deposited on the working electrode and incubated for at least 1 h, at 22 °C. The carbon surface was then blocked with 30 μL of 5% (*w*/*v*) skimmed-milk powder in PBS-tween buffer for 1 h, at 22 °C. Following the blocking step, the electrodes were rinsed with Milli-Q and dried with N_2_.

In the case of gold SPEs, the surface was first modified with 11-mercaptoundecanoic acid (MUA), followed by covalent binding to the β-conglutin protein. Briefly, the clean gold electrodes were immersed in 1 mM MUA in ethanol for 3 h, at 22 °C, and then washed with ethanol. The carboxyl group of the MUA was sequentially modified by adding an aqueous mixture of EDC (0.2 M) and NHS (50 mM) for 30 min. After washing the electrodes with Milli-Q water and drying with N_2_, 30 μL of 20 μg mL^−1^ β-conglutin protein prepared in acetate buffer, pH 5, was deposited on the working electrode and incubated for 1 h, at 22 °C. The remaining carboxyl groups were then blocked with 0.1 M ethanolamine hydrochloride, pH 8.5, for 30 min, at 22 °C. Finally, the non-modified gold was blocked with 1x Denhardt’s solution for 2 h. Following the blocking step, the electrodes were rinsed with Milli-Q and dried with N_2_. Cyclic voltammetry (CV) was performed after each step of surface functionalization, from −0.2 to 0.4 V vs. Ag in a solution of 1 mM K_3_[Fe(CN)_6_] containing 100 mM KCl, at 100 mV/s, for carbon electrodes; and from −0.1 to 0.3 V vs. Ag in a solution of 1 mM K_3_[Fe(CN)_6_] containing 100 mM KCl, at 100 mV/s, for gold electrodes (Appendix A).

#### 2.5.3. Evaluation of the Functionalized Screen-Printed Electrodes (SPEs)

For the evaluation of the functionalization of the electrodes, 30 μL of 0.4 nM of the aptamer containing biotinylated dUTPs or a non-cognate aptamer, also containing biotinylated dUTPs, in binding buffer (10 mM phosphate buffer, 135 mM NaCl, 2.5 mM KCl, and 1.5 mM MgCl_2_ pH 7.4) was incubated on the electrodes for 30 min, at 22 °C. Following washing with Milli-Q and drying with N_2_, 30 μL of 1 in 20,000 dilution of 1 mg mL^−1^ SA-polyHRP80 was added and then incubated for a further 30 min. A final washing step was carried out prior to the addition of 50 μL of TMB substrate, and 2 min later, chronoamperometry was performed as described above. Moreover, three non-cognate targets (α-conglutin, γ-conglutin, and δ-conglutin) were immobilized, as explained above, and incubated with the aptamer incorporating biotinylated dUTPs. For the data analysis, the amperometric signal obtained at 0.5 s was plotted. Triplicate measurements were performed for each concentration.

#### 2.5.4. Competition on Screen-Printed Electrodes

Individual Eppendorf tubes containing 0.4 nM of the aptamer containing biotinylated dUTPs were pre-incubated with serial dilutions of β-conglutin (25 nM–0 nM) in binding buffer (10 mM phosphate buffer, 135 mM NaCl, 2.5 mM KCl, and 1.5 mM MgCl_2_, pH 7.4) for 30 min, at 22 °C. Following incubation, 30 μL of the pre-incubated solutions was deposited on the electrode for a further 30 min. This was followed by rinsing with Milli-Q and drying with N_2_ and a subsequent addition of 30 μL of a 1 in 20,000 dilution of 1 mg mL^−1^ SA-polyHRP80, again for 30 min. After a final washing step, 50 μL of TMB substrate was added, and 2 min later, chronoamperometry was performed as described above. For the data analysis, the amperometric signal obtained at 0.5 s was used to construct the calibration curves, plotting with GraphPad Prism software and fitting to a sigmoidal 4PL model. The limit of detection (LOD) was calculated as described above.

## 3. Results and Discussion

A model system exploiting an aptamer previously selected against the anaphylactic allergen β-conglutin (β-CBA II aptamer) was used to explore the possibilities of improving the achievable detection limit via signal enhancement and potentially the aptamer affinity through increased aptamer–target hydrophobic interactions [34,35,36,37].

### 3.1. Aptamer Preparation: Incorporation of Biotinylated dNTPs

The preparation of the modified aptamer was achieved via a first amplification to produce double-stranded DNA, dsDNA, incorporating biotinylated dUTPs, followed by asymmetric amplification and finally the use of lambda exonuclease digestion for the generation of single-stranded DNA, ssDNA (Figure 1a) [38,39]. Agarose gel electrophoresis was used to visualize the amplicons produced at each stage. As can be seen in Figure 1b, the first amplification produced a high amount of dsDNA modified with biotinylated dUTPs, and asymmetric amplification, where only the unmodified forward primer was added, resulted in a mixture of dsDNA and ssDNA. The final step of digestion with lambda exonuclease resulted in the conversion of almost all the dsDNA to the ssDNA aptamer with the incorporated biotinylated dUTPs.

### 3.2. Evaluation of the Binding Affinity of the Biotinylated Aptamer

Surface Plasmon Resonance (SPR) was performed to investigate the effect of incorporation of biotinylated dUTPs on the affinity of the aptamer for its cognate target, as the incorporation of the hydrophobic biotin molecules could result in increased hydrophobic interactions between the modified aptamer and the β-conglutin. The constant dissociation (K_D_) of the modified aptamer was thus compared with the unmodified aptamer. However, as can be seen in Appendix A, the aptamer affinity was unaffected, with a similar K_D_ of 11.1 nM and 18.5 nM obtained for the unmodified and modified aptamer, respectively. Whilst in some cases, post-SELEX modification via the introduction of hydrophobic nucleotides can improve affinity, this is highly aptamer and target molecule specific, and the preferred route is to include these hydrophobic unnatural bases in the initial library used in SELEX. Indeed, we observed, again, for the β-conglutin target and aptamer that post-SELEX introduction of hydrophobic bases did not improve affinity, whilst selecting a new aptamer against the same target by using the same bases in the initial library resulted in a high-affinity aptamer. Any improvement in the detection limit achieved is thus purely due to the signal enhancement achieved via the incorporation of the biotinylated dUTPs for subsequent linking to SA-polyHRP and not in any way due to an improvement in affinity.

### 3.3. Evaluation of the Sensitivity of Biotinylated Aptamer Using an Enzyme-Linked Aptamer Assay (ELAA)

An Enzyme-Linked Aptamer Assay (ELAA) was carried out to confirm the incorporation of the biotinylated dUTPs during aptamer preparation (Figure 2a), where a range of concentrations of the modified aptamer was tested against the β-conglutin modified microtiter plate. The signal intensity was compared with that obtained from using the aptamer modified with a single biotin at the 5′ end (the modification of this aptamer in 5′ does not affect its functionality or affinity [34,35,36]). As can be seen in Figure 2b, whilst the saturating signal was the same for both aptamers, higher signals were observed with lower concentrations of the aptamer with incorporated biotinylated dUTPs, obtaining an IC50 that was four times better (0.142 nM and 0.609 nM for biotinylated dUTPs aptamer and biotinylated aptamer with a single biotin, respectively) and an enhancement of one order of magnitude of the limit of detection (LOD) of the assay (0.004 nM and 0.058 nM for the aptamer containing biotinylated dUTPs and for the biotinylated aptamer with a single biotin, respectively). Moreover, to confirm the specificity of the assay, three non-cognate targets and a non-cognate aptamer (Seq. 5) were used as controls [35]. The β-conglutin was replaced by α-conglutin, γ-conglutin, or δ-conglutin; the biotinylated dUTPs aptamer was tested against these targets; and negligible binding was observed. The non-cognate aptamer was modified in the same manner as the biotinylated dUTPs aptamer β-CBA II and tested against the β-conglutin target; again, negligible binding was observed, demonstrating the specificity of the biotinylated dUTPs β-CBA II aptamer against β- conglutin.

### 3.4. Competition Assay on Microtiter Plate: Enzyme-Linked Aptamer Assay (ELAA)

The competitive ELAA was based on the detection of β-conglutin via competition between the target immobilized on the microtiter plate and the target in the solution. With increasing concentrations of β-conglutin in solution, less aptamer is free to bind to the immobilized β-conglutin. As a result, there is a decrease in the intensity of the signal, whilst in the absence of target, the modified aptamer binds to immobilized β-conglutin (Figure 3a).

The concentrations of coating β-conglutin and the β-CBA II aptamer are the important parameters that need to be optimized in competitive binding assays, since they are the limiting factors to obtain the higher sensitivity of the assay [40], and they were thus optimized. With the aim to determine the minimal concentration of functionalized β-conglutin on the plate with the highest signal (Appendix A), different concentrations of immobilized β-conglutin were incubated with a constant amount of biotinylated aptamer (1 nM), followed by SA-polyHRP. The optimum concentration was determined to be 20 µg mL^−1^, and this concentration was used in all further experiments. To elucidate the optimal concentration of the modified aptamer (Appendix A), different concentrations of aptamer were added to the wells of a microtiter plate modified with 20 µg mL^−1^ of β-conglutin. The supernatant containing any unbound aptamer was removed and added to a fresh plate, which was also modified with 20 µg mL^−1^ of β-conglutin. By comparing both binding curves, we demonstrated the 0.4 nM aptamer to be ca. 80% of the saturating concentration, and this was confirmed by the low response obtained with the supernatant.

A calibration curve was then constructed by using these optimized concentrations and a range of concentrations of β-conglutin, resulting in a LOD of 68 pM with the aptamer with the incorporated biotinylated dUTPs, which was three orders of magnitude lower than the LOD obtained with the aptamer bearing a single biotin label (31.39 nM). This markedly lower detection limit was similar to that obtained by using more complex, multistep assays, such as Apta-PCR and Apta-RPA [37] (Figure 3b).

### 3.5. Electrochemical Detection

Having demonstrated the proof of concept by using colorimetric detection with microtiter plates, the assay was then applied to electrochemical detection, using screen-printed electrodes, with the aim of further improving the limit of detection [41] and to move to a more portable approach that could find application at the point of need. Gold and carbon screen-printed electrodes (SPEs) were evaluated as transducers for this assay.

For the carbon SPE, the approach was directly transferred from the ELAA; the β-conglutin was physically adsorbed on the surface of the electrode, and skimmed milk was used as blocking agent (Figure 4a). In the case of gold, 6-mercaptoundecanoic acid (MUA) was self-assembled on the gold surface, creating a monolayer with a terminal carboxyl group, which was subsequently used for chemical crosslinking via the formation of an amide bond with the β-conglutin (Figure 5a). Compared to the adsorption approach, in this covalent functionalization, the protein was oriented in the same way that had been employed during selection of the aptamer [35]. This more favorable orientation of the protein, together with the higher conductivity of gold as compared with carbon, should favor a higher sensitivity. Denhardt’s solution was used as a blocking agent, as it more efficiently blocked the gold from non-specific adsorption of β-conglutin [34].

In order to evaluate the successful functionalization of the electrodes with the protein, modified aptamer (0.4 nM) was added to the electrode surface, and following incubation, SA-polyHRP and TMB were added, and the subsequent reduction signal was measured by using fast chronoamperometry. As a control, electrodes without β-conglutin coating, a non-cognate aptamer, and three non-cognate targets were also tested, demonstrating the specificity of the assay (Appendix A). Finally, the competitive assay was performed by using both electrodes, and LODs of 326 fM and 7.89 fM were obtained for carbon and gold electrodes, respectively (Figure 4b and Figure 5b).

## 4. Conclusions

Signal enhancement as a means of achieving lower detection limits is of increasing interest for aptamer-based detection, where either ultrasensitive detection limits are required or the performance of low-affinity aptamers needs to be improved. Taking advantage of the inherent nucleic acid nature of aptamers, we described a generic method, applicable to all aptamers, where we incorporated biotinylated dUTPs into the aptamer structure. We evaluated the affinity of this modified aptamer by using surface plasmon resonance to explore if the affinity had been adversely affected by the presence of the biotin labels, or if the affinity had even been improved due to the enhanced hydrophobicity of the aptamer, which could facilitate improved access of the aptamer to hydrophobic moieties of the β-conglutin target. Whilst the affinity was not affected, no improvement in the affinity was observed either, and this is not entirely surprising, as SELEX had not been carried by out using an initial library modified with the biotinylated dUTPs, and future work will investigate the use of the biotinylated dUTPs in the initial library. However, whilst we observed this for the specific example of the β-CBA II aptamer with the β-conglutin target, it may not be universally applicable, and there may be other examples where aptamer affinity could be negatively affected or significantly improved, and it is assumed that this will be aptamer and target-specific.

To demonstrate the proof-of-concept, initially a colorimetric aptamer-based competitive assay was performed and a LOD of 68 pM obtained, representing a three-orders-of-magnitude improvement as compared to the LOD of 31.39 nM achieved by using the aptamer bearing a single biotin label at the 5′ terminal. The modified aptamer was then implemented in an electrochemical assay by using a screen-printed electrode, with significantly improved detection limits of 326 fM and 7.89 fM observed for carbon and gold electrodes, respectively.

This generic approach for signal enhancement is facilely implemented, is cost effective, and avoids the need for multistep assays to achieve lower detection limits. Ongoing work is focused on incorporating biotinylated dNTPs in the initial library used in SELEX, as well as in expanding the demonstrated platforms to multiplex detection and exploring the use of directly detectable electroactive labels in the aptamer sequence.

## Figures and Tables

**Figure 1 biosensors-12-00972-f001:**
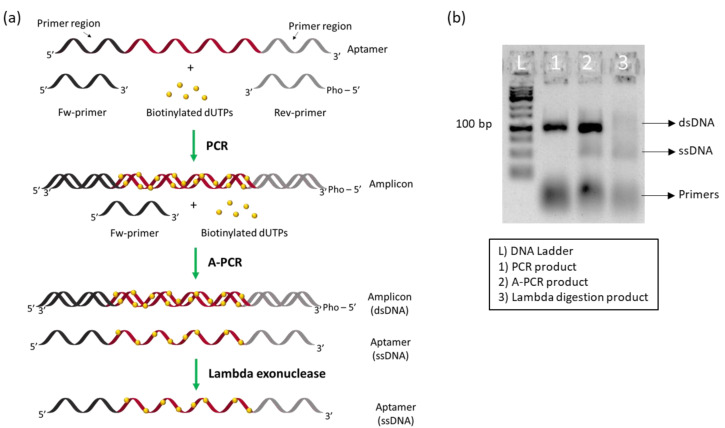
Aptamer preparation using biotinylated dUTPs: (**a**) schematic representation of the prepared aptamer; (**b**) agarose gel, showing the different steps needed for the modified aptamer.

**Figure 2 biosensors-12-00972-f002:**
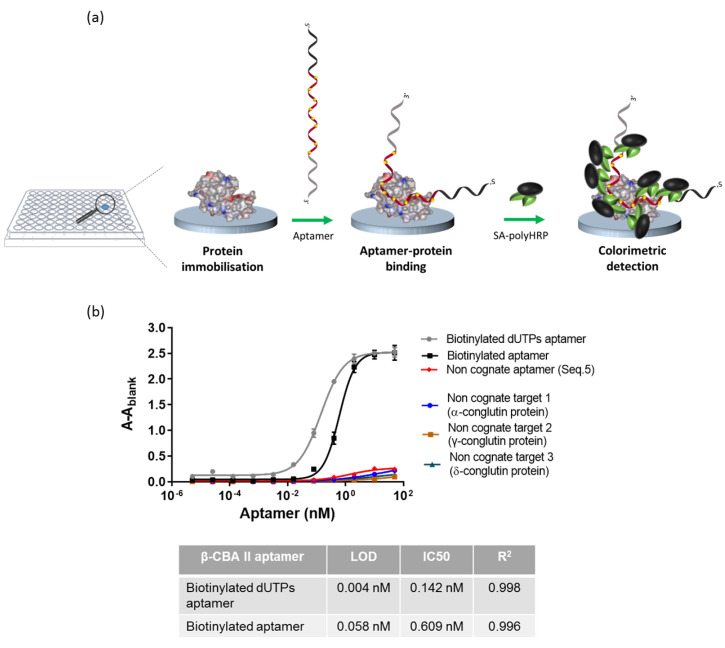
Evaluation of biotinylated aptamer, using an Enzyme-Linked Aptamer Assay (ELAA): (**a**) schematic representation of the assay; (**b**) binding curves comparing modified aptamer with biotinylated dUTPs and modified biotinylated aptamer (with a single biotin label) and control assays, using a non-cognate aptamer with biotinylated dNTPs, and the specific aptamer against three non-cognate targets.

**Figure 3 biosensors-12-00972-f003:**
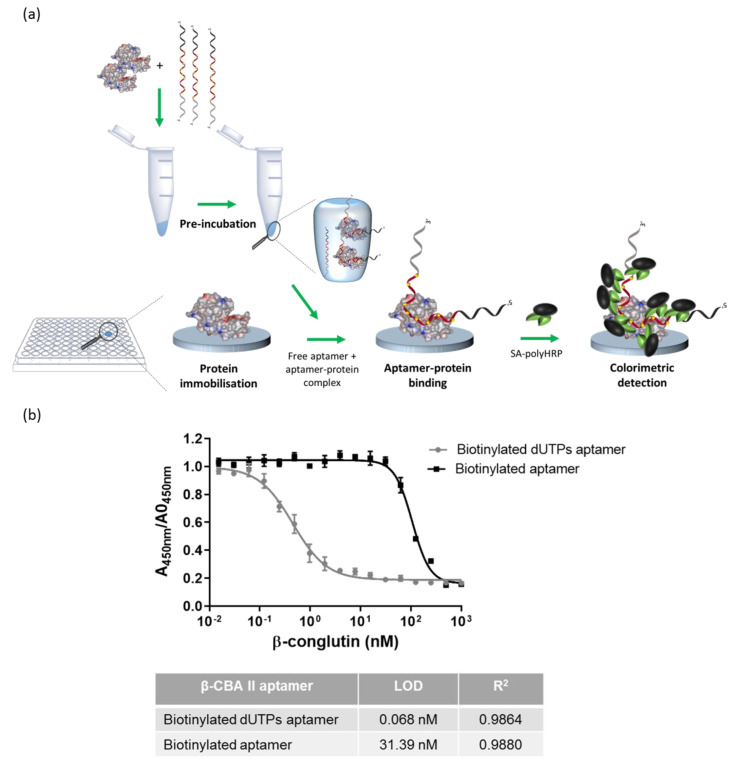
Competition assay on microtiter plate: (**a**) schematic representation of the assay; (**b**) calibration curves comparing modified biotinylated dUTPs aptamer and aptamer bearing a single biotin label (biotinylated aptamer).

**Figure 4 biosensors-12-00972-f004:**
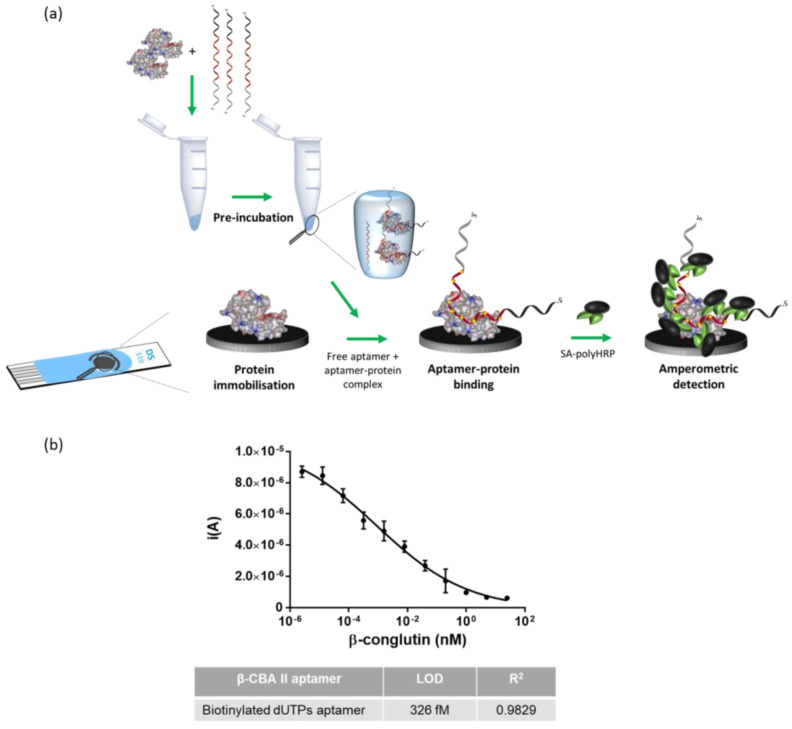
Competition assay on carbon screen-printed electrode: (**a**) schematic representation of the assay and (**b**) calibration curve.

**Figure 5 biosensors-12-00972-f005:**
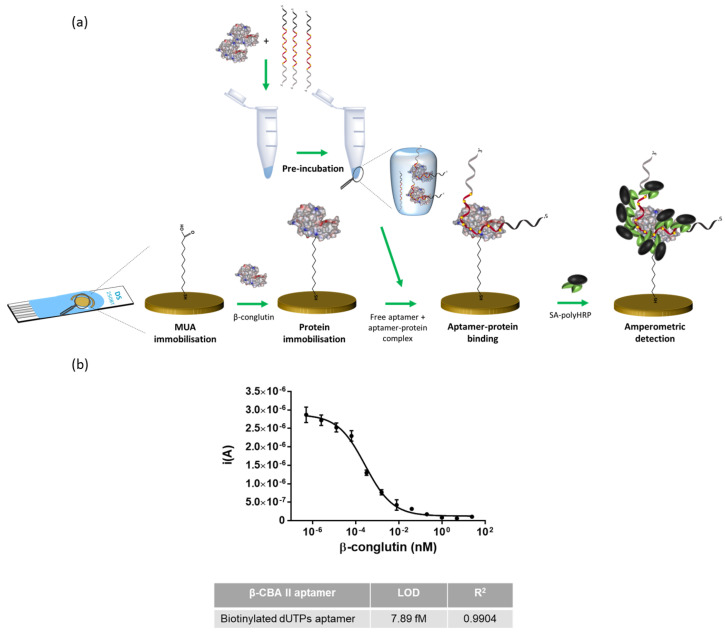
Competition assay on gold screen-printed electrode: (**a**) schematic representation of the assay and (**b**) calibration curve.

## Data Availability

Not applicable.

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
