# Peer review of "Exploiting the Nucleic Acid Nature of Aptamers for Signal Amplification"

_biosensors, 2022, doi:10.3390/bios12110972_

Round 1

Reviewer 1 Report

The authors have reported the advantage of the inherent nucleic acid nature of aptamers to enhance sensitivity in a rapid and facile assay format. The aptamer was generated using biotinylated dUTPs was then applied in a colorimetric competitive enzyme linked oligonucleotide assay and was used with screen-printed electrodes in electrochemical detection.

This work can be reconsidered to be published after major revision and only if controls asked are realised. The comments are listed as follows:

1- The study is interesting and novelty, however there is a lack of important and primordial controls to verify the veracity of the interpretations. A non cognate aptamer (scramble sequence) should be used as control. It should be prepared using biotinylated dUTPs in the same manner than aptamer β-CBA II aptamer. After, the evaluation of the binding affinity of the biotinylated aptamer should be realised also for the scramble aptamer. In addition, the competitive ELAA was based on the detection of β-conglutin via competition between the target immobilised on the microtiter plate and the target in the solution has to be realised with both non cognate aptamer and non cognate target to check the specificity of the assay. The same experiments control (aptamer and target non cognate) must be performed for competition assay on carbon screen-printed electrode and gold screen-printed electrode.

2- Figure 2 should be revised. The curves do not start at the same absorbance value, to avoid this phenomenon and to avoid overestimating the effect of biotinylated dUTPs aptamer, the y-axis should represent A-Ablank (Ablank : well with only reagents). This is to compare exactly biotinylated dUTPs aptamer and biotinylated aptamer.

3- Evaluation of the sensitivity of biotinylated aptamer using an Enzyme Linked Aptamer Assay

(ELAA). This paragraph needs to be improved.

4- Figure 3b: all the curves must be on the same figure with appropriate controls explained above.

5- Line 401: This method is not generic because is not applicable to other aptamers and targets. I think that the authors have to prove the generalisability by making other experiments on other aptamer and targets.

6- In the supplementary materials, the authors have to put experimental conditions in the figure legends.

Reviewer 2 Report

This work by Jauset-Rubio, M et al. describes the use of biotin modified aptamers to increase the detection sensitivity of DNA aptamers. While the work presented is significant and done with proper controls, there are some outstanding minor concerns.

·      Why is the maximum signal not increasing in Fig2? Presumably if aptamer has more biotin and hence more SA-HRP, maximum signal should increase.

·      Are the authors correcting for saturation of detectors while measuring absorbance?

·      Do authors think other small molecule incorporated in the DNA can increase the sensitivity as well or is the effect limited to biotin?

Round 2

Reviewer 1 Report

The authors have reported the advantage of the inherent nucleic acid nature of aptamers to enhance sensitivity in a rapid and facile assay format. The aptamer was generated using biotinylated dUTPs was then applied in a colorimetric competitive enzyme linked oligonucleotide assay and was used with screen-printed electrodes in electrochemical detection.

1)      The authors have considered the suggested advice for previous comments (2, 3, 4, 5 and 6).

2)      For comment 1, the authors have added the experiments necessary to validate the results. We can just deplore that they don’t put the controls in Figure 4 and 5 like for figure 2. In addition, the authors have chosen not to carry out controls for Competition assay on microtiter plate: Enzyme Linked Aptamer Assay (ELAA), they have not justified this choice but we can anticipate that the competition doesn’t change the fact that the aptamer is specific. But the ideal would have been to prove that the competition was impossible with a non cognate target.

3)      The authors should be change the title of 3.3 section and added the term of specificity.

3.3 Evaluation of the specificity and sensitivity of biotinylated aptamer using an Enzyme Linked Aptamer Assay (ELAA)

To conclude, this work can be published in present form.